# Aberrant Signaling Pathways in Sinonasal Intestinal-Type Adenocarcinoma

**DOI:** 10.3390/cancers13195022

**Published:** 2021-10-07

**Authors:** Cristina Riobello, Paula Sánchez-Fernández, Virginia N. Cabal, Rocío García-Marín, Laura Suárez-Fernández, Blanca Vivanco, Verónica Blanco-Lorenzo, César Álvarez Marcos, Fernando López, José Luis Llorente, Mario A. Hermsen

**Affiliations:** 1Department of Head and Neck Oncology, Instituto de Investigación Sanitaria del Principado de Asturias, 33011 Oviedo, Spain; cristinarisu15@gmail.com (C.R.); vircabal@hotmail.com (V.N.C.); rociogm220879@hotmail.com (R.G.-M.); laura_quillo@hotmail.com (L.S.-F.); 2Department of Otolaryngology, Hospital Universitario Central de Asturias, 33011 Oviedo, Spain; psfernan90@hotmail.com (P.S.-F.); caalvarez@uniovi.es (C.Á.M.); flopez_1981@yahoo.es (F.L.); llorentejlx@gmail.com (J.L.L.); 3Department Pathology, Hospital Universitario Central de Asturias, 33011 Oviedo, Spain; vivancoblanca@gmail.com (B.V.); vblancolorenzo@gmail.com (V.B.-L.)

**Keywords:** sinonasal cancer, intestinal-type adenocarcinoma, next generation sequencing, gene mutations, signaling pathways

## Abstract

**Simple Summary:**

The majority of patients with sinonasal intestinal-type adenocarcinoma are wood and leather workers. However, the genetic changes that lead to these tumors are not well known. We analyzed 50 tumors for mutations in a set of 120 genes that may be involved in causing this cancer type. We found that 72% of cases carried mutations in genes that are active in Wnt, DNA damage response, MAPK or PI3K signaling pathways. Pathway activation was not related to mutations of genes in these pathways, except for nuclear β-catenin expression to Wnt pathway mutation. No specific gene mutation, mutated pathway, nor pathway activity level was associated with histological subtype, clinical data or survival. Finally, none of the identified mutated genes occurred in such frequency as to be considered a characteristic genetic feature of sinonasal intestinal-type adenocarcinoma.

**Abstract:**

Sinonasal intestinal-type adenocarcinoma (ITAC) is strongly related to occupational exposure to wood and leather dust, however, little is known on the genetic alterations involved in tumor development and progression. The aim of this study was to identify tumorigenic signaling pathways affected by gene mutations and their relation to clinical features. We applied whole exome sequencing of 120 cancer-related genes in 50 ITACs and analyzed the signaling activity of four specific pathways frequently affected by mutations. Genes involved in DNA damage response showed somatic mutations in 30% of cases, including four tumors that also harbored germline mutations. Genes in Wnt, MAPK and PI3K pathways harbored mutations in 20%, 20% and 24% of cases, respectively. Mutations and copy number gains in receptor tyrosine kinases possibly affecting MAPK and PI3K pathways occurred in 44% of cases. Expression of key pathway proteins showed no correlation to mutations in these pathways, except for nuclear β-catenin and APC/CTNNB1 mutation. No specific gene mutation, mutated pathway, nor pathway activity level showed correlation to clinical data or survival. In addition, a similar mutational profile was observed among histological subtypes. The wide spectrum of gene mutations suggests that ITAC is a genetically heterogeneous without specific characterizing gene mutations.

## 1. Introduction

Epithelial tumors are the most frequent malignancies in the sinonasal region, and intestinal-type adenocarcinoma (ITAC) is the second most frequent subtype, constituting 10–20% of all sinonasal tumors, depending on the geographical region [1]. ITAC almost exclusively arises in the olfactory mucosa of the ethmoid sinus. It is etiologically related to occupational exposure to several industrial compounds, especially wood and leather dust, and for this reason is considered a professional disease in various European countries. There are four histological subtypes, papillary, colonic, solid and mucinous, while mixed histologies also occur [2,3]. Histologically, ITAC mimics adenoma and adenocarcinoma from the intestinal mucosa; sometimes it even mimics the normal intestinal mucosa histology hence the denomination ‘intestinal-type’ [4]. The typical immunophenotype of ITAC includes positive staining for pancytokeratins, EMA and markers of intestinal differentiation such as cytokeratin 20, CDX2, villin, SATB2 and MUC2 [3,4,5]. Although CK7 staining is indicative of ITAC, there are no markers that clearly distinguish ITAC from metastatic colorectal adenocarcinomas (CRC), so exclusion of a primary colorectal tumor is necessary in the diagnosis [4].

Due to the lack of symptoms in early stages, these tumors are commonly detected when already invading into surrounding areas. Local recurrence and intracranial invasion are the most common causes of death. The standard treatment is surgical resection, with a minimally invasive endoscopic approach when possible [1,6]. Postoperative radiation is commonly administred given the high recurrence rate of this tumor type, and advanced stage tumors are also treated with neoadjuvant chemotherapy [7]. Recurrence occurs in up to 50% of cases, whereas lymph-node or distant metastasis is infrequent, in 8–13% of cases. The 5-year overall survival rate is approximately 60% [1,8].

In spite of our growing knowledge on the genetic aberrations in ITAC, the major signaling pathways involved in tumorigenesis are still unknown. The published data so far have produced heterogeneous results, with variable frequencies, for instance, TP53 in 18–86% of cases [9,10,11,12,13] KRAS 0–50% [10,13,14,15,16,17,18,19,20] and APC 0–37% [10,14,15]. There is more agreement on genes such as EGFR, HRAS, BRAF and CTNNB1 (coding β-catenin) that are apparently not involved in ITAC, as reported mutations are in the range of 0–6% of cases [10,12,14,15,17,18,19,20,21].

Microarray CGH studies have pointed out short homozygous deletions that affect classical cancer genes such as APC (5q13), PTEN (10q23), CDH1 (16q22) and TP53 (17p13) and also chromosome 7 gains that comprise EGFR and MET [22,23,24,25,26]. Gene copy number analysis using MLPA and FISH have confirmed the gains of these latter two genes and also indicated frequent gain of FGFR1 on chromosome 8p11 [27,28,29]. 

The effects of gene mutations and copy number abnormalities on protein expression of cellular pathways have been validated via immunohistochemical studies. P53 overexpression has been observed in 50–70% of ITACs, nuclear b-catenin (coded by CTNNB1) in 30–50%, EGFR in 20–30% and cMET in 64% [11,17,27,28,30].

From all these studies it appears that ITAC shares some genetic features with histologically similar CRC and lung adenocarcinoma, such as Wnt pathway alterations (APC, CTNNB1) affecting the cell differentiation and receptor tyrosine kinases (EGFR, cMET, FGFR1) with possible downstream effects on MAPK and PI3K pathways. On the other hand, the much lower proportion of cases affected by these alterations suggest that additional genetic events must play a role in ITAC development. Recently, our group published a next-generation sequencing (NGS) study on 48 ITACs that indicated several gene mutations as targets for new personalized therapies, confirming recurrent mutations in APC and KRAS and reporting novel mutations in ATM, BRCA1, NF1 and LRP1B [31]. The present study extended the previous NGS study to 50 cases and focused on the type and distribution of the mutations, as well as on possible germline mutations and gene copy number alterations (CNAs). Our aim was to investigate the signaling activity of four pathways frequently affected by gene mutations and the relation to clinical features.

## 2. Material and Methods

### 2.1. Patients and Tissue Samples

Fresh frozen primary tumor samples were collected from 50 ITAC patients. In 29 of these cases also peripheral blood samples were available. All experimental protocols were approved by and carried out according to the Institutional Ethics Committee of the Hospital Universitario Central de Asturias and by the Regional CEIC from Principado de Asturias (approval number: 83/17 for project PI17/00763 and 07/16 for project CICPF16008HERM). Informed consent was obtained from all patients. All patients were male, 49–88 years of age (mean 69 years), and all but one with professional exposure to wood dust. Three cases were papillary subtype, 29 colonic, 6 solid and 12 mucinous subtype. Follow-up ranged between 0 and 187 months (median 50). Thirty-one (62%) patients developed local recurrence, distant metastasis appeared in 6 (12%), and 22 (44%) cases died of disease. The complete clinical characteristics of all 50 ITAC patients are summarized in Table 1.

### 2.2. Next-Generation Sequencing (NGS)

Tumor DNA from tissue samples was extracted using a Qiagen Tissue Extraction Kit (Qiagen GmbH, Hilden, Germany), and normal germline DNA was isolated from peripheral blood with a High Pure PCR Template Preparation Kit (Roche Diagnostics GmbH, Mannheim, Germany). Twenty-nine tumor and matched germline DNAs as well as 21 DNAs of only tumor cases were sequenced using a NGS panel of 120 cancer-related genes: AKT, AKT1, AKT3, ALK, APC, AR, ARAF, ATM, ATR, AURKA, BAP1, BCL2L1, BCR-ABL1, BCR-JAK2, BRAF, BRCA1, BRCA2, BRD4, CBL, CCND1, CCNE1, CDK4, CDK6, CDKN1A, CDKN1B, CDKN2A, CDKN2B, CDKN2C, COL1A1-PDGFRB, CRLF2, CSF1R, CSF3R, CTNNB1, DDR2, DNMT3A, EGFR, EPHA2, ERBB2, ERBB3, ERBB4, ERCC1, ERS1, EZH2, FBW7, FBXW7, FGFR1, FGFR2, FGFR3, FLT3, FOXA1, FOXL2, FOXP1, GNA11, GNAQ, HGF, HRAS, IDH1, IDH2, IGF1, IGF1R, IGF2, IL10, IL7R, INPP4B, JAK1, JAK2, JAK3, KIT, KRAS, LRP1B, MAP2K1, MAP2K2, MAP2K4, MCL1, MDM2, MET, MGMT, MITF, MLL, MPL, mTOR, MYCN, MYD88, NF1, NF2, NFKB1, NFKB2, NOTCH1, NOTCH2, NOTCH3, NPM1, NRAS, NTRK1, PALB2, PDGFRA, PIK3CA, PIK3R1, PIK3R2,PML-RARA, PTCH1, PTEN, RAC1, RAF1, RB1, RET, RET-PTC1, ROS1, SH2B3, SMO, SOCS1, STAG2, STK11, TMPRSS2-ERG, TMPRSS2-ETV1, TSC1, TSC2 [31]. A SureSelect QXT Target Enrichment Kit for Ilumina Multiplexed Sequencing was applied following the manufacturer’s instructions (Protocol Version D0, November 2015, Agilent Technologies, Santa Clara, CA, USA). Twenty-five nanograms of genomic DNA quantified using a Qubit HS dsDNA kit (Life Technologies, Carlsbad, CA, USA) was fragmented and adaptors were added in a single enzymatic step. The adaptor-tagged DNA library was purified and amplified. Next, 750 ng of each library was hybridized using SureSelect QXT capture library. The resulting libraries were recovered using Dynabeads MyOne Streptavidin T1 magnetic beads (Life Technologies, Carlsbad, CA, USA), and a post-capture PCR amplification and indexing of the samples was carried out. After each step, the purification step was performed with AMPure XP beads (Beckman Coulter, Brea, CA, USA) to remove short fragments such as adapter dimers. The quality of the libraries was assessed on a Bioanalyzer High Sensitivity DNA chip (Agilent Technologies, Santa Clara, CA, USA). Based on DNA concentration and average fragment size, libraries were normalized to an equal concentration, 5 nM, and pooled by equal volume in 16-plex pools. Sequencing pools were then sequenced in a MiSeq system (Ilumina Inc. San Diego, CA, USA). The average coverage of the sequencing was always more than 300X reads, with a mean around 500X.

### 2.3. Bioinformatic Analysis

Sequencing raw data was processed using the bioinformatics software HD Genome One (DREAMgenics, Oviedo, Spain), certified with IVD/CE-marking. The pipeline included quality control and alignment, somatic variant calling (when normal DNA sample was available), variant annotation and copy number variantion (CNV) and copy number-neutral loss of heterozygosity (CN-LOH) identification. The detection of CNVs was performed using a modified version of the exome2cnv algorithm [32], incorporating a combination of read depth and allelic imbalance computations for copy number assessment. For each sample, the algorithm employs a pool of samples, prepared with the same library, as background for the detection of CNVs. The datasets generated in the study are in the process of being deposited in a publicly available repository. Manual curation of the annotated variants from the 50 ITAC tumor samples was performed in order to select only the probably damaging changes affecting the tumors. For this, we used different strategies for the 29 tumor/germline matched and the 21 tumor-only cases. 

To select somatic mutations with a pathogenic impact on the 29 tumor/germline matched cases, we filtered out all variants with allele frequency >5% in the normal population, all silent mutations and all variants with a tumor frequency <10% of the total reads. We considered as true relevant somatic variants those that appeared in the tumor sample but not in the normal germline sample of the patient, as well as changes from a heterozygous germline variant to a homozygous tumor variant. To evaluate the mutational spectrum of the tumors and the proportions of the different nucleotide changes (A > T, A > C, A > G, T > A, T > C, T > G, C > A, C > T, C > G, G > A, G > T, G > C), we filtered and took into account all the somatic and LOH variants (high frequency biallelic genetic inactivation of tumor suppressor genes from heterozygous variants in the germline sample).

For the 21 tumor-only cases we filtered out the variants with allele frequency >1% in the normal population, all silent mutations and all variants with frequency <10% of the total reads in the tumor sample. This threshold of >10% of the variant allele frequency in the tumor was to select those variants with a relevant impact on tumor biology, while variants with a tumor allele frequency <10% were assumed to be passenger mutations. Next, we took from this selection only those non-synonymous changes that are registered on COSMIC or ICGC databases or have a known protein effect. 

### 2.4. Immunohistochemistry

Tissue microarray (TMA) blocks were prepared from formalin fixed, paraffin embedded tumor tissues using Beecher Tissue Microarray (Beecher Instruments, Silver Spring, MD, USA). TMA blocks contained three 1 mm cores from different areas of each tumor. Each block included normal sinonasal mucosa samples as internal control. Tumors not included in the TMAs were stained separately. Immunohistochemistry was performed on an automatic staining workstation (Dako Autostainer Plus; DakoCytomation, Glostrup, Denmark) with antigen retrieval using EnVision FLEX + Mouse (DakoCytomation, Glostrup, Denmark) for 20 min. The following antibodies were applied in order to study the effect of the genetic variants found: p-ERK 1:500 (rabbit anti-phospho-p44/42 MAPK Erk1/2 Thr202/Tyr204 clone D13.14.4E, Cell Signaling, MA, USA), p-mTOR 1:100 (rabbit anti-phospho-mTOR Ser2448 clone 49F9, Cell Signaling, MA, USA), CTNNB1 1:200 (Mouse anti-β-catenin clone 14/Beta-Catenin, BD Transduction Laboratories, CA, USA) and PARP1 1:100 (mouse anti-PARP1 clone F-2, Santa Cruz Biotechnology, Santa Cruz, CA, USA).

Immunostaining for nuclear and/or cytoplasmic pMAPK and pmTOR was scored on a four-tiered scale for intensity (0 absent, 1 weak, 2 moderate, 3 strong) and percentage of positive tumor cells (1–25%, 26–50%, 51–75% and 76–100%), creating a final 4 level score by multiplicating intensity and precentage scores: (0, 1 (score 1–4), 2 (score 5–8), and 3 (score 9–12)). Nuclear stain of β-catenin was evaluated semiquantitatively and scored as: 0 (0% positive cells), 1 (1–25% positive cells), 2 (25–50% positive cells) and 3 (>50% positive cells) [33]. Nuclear PARP1 expression was scored according to the following multiplicative score: percentage of positive cells in a score from 0 to 6 (0 = 0%, 1 = 1–4%, 2 = 5–19%, 3 = 20–39%, 4 = 40–59%, 5 = 60–79% and 6 = 80–100%), while the intensity was scored on a four-tiered scale (0 absent; 1 weak; 2 moderate; and 3 strong). Resultant scores were classified as low (0–9) or high (9–18) [34].

### 2.5. Statistical Analysis

Correlations between the immunohistochemical stainings, pathway-related gene mutations and clinico-pathological variables were analyzed using IBM SPSS Statistics 25.0 (SPSS Inc., Chicago, IL, USA), using Pearson chi-square and Fisher’s exact test. Kaplan-Meier analysis was performed for estimation of survival, comparing distributions of survival through the logarithmic range test (log-rank test). Values of *p* < 0.05 were considered significant.

## 3. Results and Discussion

To identify somatic alterations that may provide new insights on ITAC tumorigenesis, we analyzed 50 tumors using a NGS panel consisting of 120 cancer-related genes. In 29 cases we were able to co-analyze germline DNA, allowing only somatic sequence variants to be exclusively selected. In 21 cases where only tumor DNA was sequenced the somatic status could not be verified, however, using stringent bioinformatic filtering we selected those variants that most likely are pathogenic. Forty-eight tumors of the present series were analyzed in our previous study that focused on mutations with therapeutic relevance, specifically the proven somatic mutations in 27/48 cases where tumor and corresponding germline DNA was available. In the present study we investigated the type and distribution of mutations in all 50 tumors, focusing on four tumorigenic signaling pathways affected by gene mutations and their relation to clinical features. In addition, we analyzed germline mutations and CNAs in the 29 tumor/germline matched cases.

### 3.1. Recurrent Gene Mutations

In the 29 tumor/germline matched cohort we identified 2.2 somatic variants per case, in the range 0–10. Seven tumors (24%) did not show any somatic mutation. Among the 120 genes sequenced, 36 showed one or more somatic mutations, in decreasing order being synonym, missense, nonsense, splice, frameshift and in-frame deletion and insertion mutations (Figure 1, Appendix A). PIK3CA and APC were the most frequently mutated genes, each observed in 5 tumors (17%), followed by ATM in 4 tumors, (14%). LRP1B, KRAS, ERBB3, BRCA1, NF1 and AR showed mutations in 3 cases each (10%), while NOTCH2 and CTNNB1 mutations were found in 2 tumors (6%). Twenty-five genes, KMT2A, BRAF, MAP2K1, NOTCH3, JAK3, CDKN2A, CDKN1B, PIK3R2, MTOR, SMO, FOX1A, FOXL2, FOXP1, ERBB2, ERBB4, KIT, DDR2, ESR1, PDGFRA, ROS1, DNMT3A, EZH2, IDH1 and BRCA2, were mutated only in one case, illustrating the heterogeneous mutational profile of ITAC. The most frequent nucleotide changes were G > A (19%), C > T (15%) and A > G (13%) (Figure 1). The frequent G > A transversions are in agreement with previous findings that G > A transversions in TP53 and KRAS genes are related to a chronic inflammatory environment, presumably induced by exposure to wood dust [11,35].

In the 21 tumor-only group we found 2.8 possibly somatic variants per case (Appendix A), in the range 0–6. Two tumors did not carry any mutation. AR was the most frequently mutated gene in 7 tumors (33%). ATM, LRP1B, BRCA1 and TSC2 mutations occurred in 4 tumors (19%), APC, KRAS and EPHA2 in 3 tumors (14%) and ERBB2, NTRK1 and JAK3 in 2 cases (9%). Finally, 17 genes, FLT3, NF1, NOTCH1, NOTCH2, NOTCH3, AKT1, MTOR, PDGFRA, ROS1, MET, FOXP1, IL7R, CSF1R, KMT2A, SMO and PCH1, showed mutation in one case each. Several of the mutated genes in this series were found to mirror exactly the variants found in the 29 tumor/germline matched series: AR (c.234_239delGCAGCA, c.303_308dupGCAGCA and c.237_239delGCA), BRCA1 (c.4039A>G), JAK3 (c.2164G>A), KRAS (c.35G>A) and TCS2 (c.5383C>T). These variants are known from COMIC and/or ICGC databases. On the other hand, 19 mutations were not published before in the public databases (Appendix A). As in the 21 tumor-only cohort there was no matched germline analysis, we have no information on gene copy number alterations.

Taking together the two cohorts, our NGS analysis uncovered the following genes as being most frequently affected by mutation: AR (20%), ATM and APC (16%), LRP1B and BRCA1 (14%), KRAS (12%) and PIK3CA (10%) (see Figure 2). Although none of the mutated genes affect a majority of tumors, they do possibly play a role in the biology of ITAC. Interestingly, the AR gene (androgen receptor) is located on chromosome X, supporting the theory that defects in genes on this chromosome could contribute to the male predominance in the development of ITAC apart from the occupational exposure [36]. It may be interesting to further study the role of this gene in ITAC. 

### 3.2. Genetic Profile of ITAC

Previous single gene studies on ITAC that investigated possible mutations in APC, TP53, KRAS and BRAF concluded that not only histologically but also genetically ITAC resembles CRC, in spite of absence of truncating APC and only very few BRAF mutations [14,15,22,23,25,37]. Recently, a NGS study by Sjostedt et al. on 19 ITAC cases reported 37% mutations in APC (16% were truncating), 58% in TP53 and 10% in KRAS, as well as 37% inactivating frameshift mutations in CHD2, and possibly (the somatic status of these variants was not shown) also 37% missense mutations in KMT2C and 21% in SDHA, and concluded ITAC to be genetically similar to CRC [10]. Unfortunately, as our NGS panel was originally designed to analyze clinically actionable gene mutations, it does not include TP53, CHD2, KMT2C and SDHA, but our data confirm that ITAC carries truncating APC and activating KRAS and BRAF mutations, however in a much lower proportion of cases than CRC. Also similar to CRC are the frequent mutations in ATM, BRCA1, LRP1B and PIK3CA that we presented in our previous study [31].

Comparing with other cancer types in the sinonasal area our data indicate a unique and heterogeneous genetic profile for ITAC. Sinonasal squamous cell carcinoma have been shown to carry frequent EGFR [38,39,40,41], whereas in our series of 50 ITACs, we did not find EGFR mutations but instead did in ERBB2, ERBB3 and ERBB4. Also recurrent in SNSCC are mutations and homozygous deletions affecting CDKN2A [42], while in contrast, we found only one case with mutated CDKN2A and no homozygous deletions. IDH2 mutations occur in high frequency in sinonasal undifferentiated carcinoma [43,44,45] but did not appear in our series of ITACs. Genes SMARCB1, SMARCA4 and NUT1, which characterize other subsets of sinonasal undifferentiated carcinoma [4,36], were not included in our 120 gene panel, but a previous immunohistochemical study including ITAC yielded no aberrations for these genes [45].

### 3.3. Germline Mutations and Copy Number Alterations

Our data revealed evidence for 11 different germline mutations in 8 of the 29 tumor/germline matched cases. All had an allele frequency of 0.37–0.53 in the blood DNA sample, while in the tumor DNA sample allele frequencies were between 0.76–0.95, indicative of a loss of the second allele. Affected genes included ATM, BRCA1, BRCA2, LRP1B, FOXA1, NOTCH3, JAK3, ERBB3 and DDR2. Appendix A gives a complete description of the variants. To our knowledge, this is the first report on hereditary germline mutations in ITAC. Aiming to gain more insight into the possible biological role of germline mutation, we checked the clinical records of the affected patients for first-grade relatives with sinonasal or other types of cancer, however, none were found. Nonetheless, it will be interesting to analyze hereditary tumorigenic germline mutations in ITAC patients in a further study.

A total of 394 CNVs were detected in the 29 tumor/germline matched cohort with a range of 0–48 alterations per case (106 gains, 255 losses and 33 copy-neutral losses of heterozygosity). However, despite the bioinformatic standardization and cut-offs, NGS using panels with a small number of genes do not represent the whole genome and have a limited reliablity to detect CNVs. In addition, possible admixture of the tumor samples with normal cells complicates setting fixed thresholds for copy number alterations. Therefore, we decided not to take into account deletions and gains of one copy (i.e., copy numbers 1 and 3). Homozygous deletions (0 copies) were not observed. Appendix A shows the copy number gains of 4 or more copies, including MET (20 copies), NRAS (10 copies), ATM (8 copies), CDK6 and CTNNB1 (6 copies), and FLT3, ERBB2, BRCA1 and HGF (5 copies). Of these gene amplifications, only MET has been studied previously, but found in 0/72 ITACs [28]. Our data confirm this study and microarray CGH studies in that ITAC does not carry frequent or recurrent gene amplifications. In this study the only recurring amplifications concerned FGFR1 in 3 cases (10%) and CTNNB1, MET, CDK6 and CCND1 in 2 cases (7%). Often these gains affected several genes in the same chromosomal region, making it difficult to deduce their biological importance as drivers of tumorigenesis. However, we noted a large number of copy number gains in genes coding for receptor tyrosine kinases (RTKs) (see paragraph below). Finally, 8/29 (27%) cases did not show any CNV, suggesting a more stable diploid chromosomal profile, as has been described in ITAC previously using microarray CGH analysis [23,24,25,26].

### 3.4. Altered Cell Signaling Pathways

When grouping mutations according to the signaling pathways in which they play a role, we found 72% of tumors affected by gene defects in Wnt (APC and CTNNB1), DNA-damage reponse (ATM, BRCA1, BRCA2), MAPK (KRAS, BRAF, NF1 and MAP2K1) and/or PI3K (PIK3CA, PIK3R2, AKT1, MTOR and TSC2) pathways, the latter two also by copy number gains in RTKs (Appendix A). Therefore, we analyzed the signaling activity of these four pathways by proxy of immunohistochemical staining of key pathway proteins and correlated the results to the status of genetic aberrations and to clinico-pathological and follow-up data. 

#### 3.4.1. Wnt Pathway

APC and CTNNB1 mutations affecting the Wnt signaling pathway occurred in 7 of 29 tumor/germline matched cases (5 APC and 2 CTNNB1) and 3 of 21 tumor-only cases (3 APC and 0 CTNNB1), in total 10 of 50 (20%), including 8 colonic and 2 mucinous type ITACs. All APC mutations were in exon 16 and concerned either nonsense or frameshift, leading to protein truncation that affects the distal region of the Apc protein. The two CTNNB1 mutations involved one misssense and one inframe deletion. In addition, we found two cases with CTNNB1 copy number gains. Nuclear β-catenin immunostaining (Figure 3) as proxy for Wnt pathway activation was found in 26/50 (52%) tumors, with 13 cases scoring 1+, 5 cases 2+ and 8 cases 3+ (Figure 4). All 10 Wnt pathway mutated cases showed nuclear staining versus 16/40 (40%) of the non-mutated cases (Fisher Chi2, *p* = 0.001). Three of 10 (30%) Wnt-mutated versus 27/40 (68%) Wnt-wildtype cases developed recurrences (Fisher Chi2 *p* = 0.067) and this was reflected in a better 5-year disease-free survival (73% and 33%, respectively, log rank 1.880, *p* = 0.170).

Adenomatous polyposis coli (APC) constitutes the ‘key’ tumor suppressor gene involved in the development of CRC, both in sporadic tumors and familial adenomatous polyposis syndrome (FAP) [46]. The role of the Apc protein consists of the binding to ß-catenin for its ubiquitination (Appendix A), so the loss of function of the gene leads to an accumulation of ß-catenin in the nucleus and cytoplasm, constitutively activating the Wnt signaling pathway, and upregulating of the transcription of important oncogenes like MYC and CCND1 [47,48].

APC mutations often co-occur with KRAS and TP53 mutations in CRC, although additional driver genes are needed for tumor development [49]. Given the histological similarity to CRC, the Wnt pathway has been studied in ITAC previously, but findings indicated absence of gene mutations in APC and CTNNB1 [14,15]. However, nuclear ß-catenin expression was found in up to 31% (26/83) of tumors, being an independent prognosticator of poor clinical outcome [30]. A recent NGS analysis revealed 16% (3/19) of tumors with protein truncating APC mutations and 4 more tumors with missense mutations whose somatic status and pathogenic relevance is not clear [10]. In our series of 50 cases, we found the same percentage of 16% APC mutations and all were truncating. Four mutations affected the β-catenin binding and downregulation domain (1342–2075aa). Our analysis also identified two cases with oncogenic mutations in CTNNB1 (p.Lys335Ile and p.Ser45del), described previously in COSMIC and ICGC databases in colon, liver and kidney carcinomas.

Our findings of Wnt mutations in 20% and Wnt pathway activation (nuclear β-catenin immunostaining) in 65% of cases suggest that Wnt pathway activation is more important in ITAC level than previously thought, possibly caused by molecular-genetic alterations additional to APC and CTNNB1 mutation. Promising specific therapies targeting the Wnt pathway in phase I clinical trials include v-ATPase inhibitors, such as bafilomycin and concanamycin that inhibit v-ATPase and have an anti-proliferative effect in xenograft and animal models of CRC [50], and porcupine inhibitors that block Wnt transport to the extracellular membrane, thus preventing excessive production of β-catenin [51]. Such therapies might also be applied in ITAC.

#### 3.4.2. DNA Damage Response Pathway

Three genes involved in the DNA damage response mechanisms *(BRCA1, BRCA2* and *ATM)* together carried 18 mutations in 16 of 50 (32%) tumors, 7 of the 29 tumor/germline matched cases and 9 of the 21 tumor-only cases. Affected ITAC cases were 1 papillary, 10 colonic, 3 solid and 2 mucinous subtype. Seven of 8 ATM mutations were missense mutations and one was a splice mutation. One of these cases showed co-occurring gene amplification. Of *7 BRCA1* mutations, 5 were missense mutations and 2 were frameshift mutations. With regard to *BRCA2,* we found one nonsense and one missense mutation. PARP1 immunostaining (Figure 3) indicating DNA damage response activity [34] was high in 30/50 (60%) cases and low in 20/50 (40%) cases (Figure 4). There was no correlation between low/high PARP1 expression and alterations in DNA damage response genes (Fisher Chi2, *p* = 1.000), and neither DNA damage response genes nor *PARP1* expression was associated to clinical or survival data.

The ATM tumor suppressor gene encodes a PI3K-related serine/threonine protein kinase (PIKK) with a central role in the repair of DNA double-strand breaks (Appendix A). Germline mutations affecting the action of the Atm protein are throught to increase cancer susceptibility, while somatic ATM mutations are among the most frequently affected genes in sporadic cancer [52]. The high proportion (32%) of ITACs affected by ATM, BRCA1 and BRCA2 mutations indicates an important role for the DNA damage response pathway in ITAC carcinogenesis and may be related to its etiology of exposure to carcinogenic wood dust particles. 

PARP inhibitors have shown clinical benefit in breast and ovary cancers that lack double-strand DNA repair, like ATM, BRCA1 and BRCA2 loss of expression tumors [52,53]. Therefore, a relevant proportion of ITAC patients could benefit from these therapies.

#### 3.4.3. MAPK Signaling Pathway

In 11 of 50 (22%) cases (7 in 29 tumor/germline matched and 4 in 21 tumor-only cases), we found recurrent mutations in MAPK signaling genes KRAS, BRAF, NF1 and MAP2K1, concerning 1 papillary, 9 colonic and 1 mucinous subtypes of ITAC. Most frequent was KRAS with 6 mutations affecting the hotspot codons 12 and 13. Four tumors carried truncating mutations in NF1, two nonsense and two frameshift alteration. The other two somatic mutations were in BRAF and MAP2K1; furthermore, we detected gene copy number gains of NRAS and BRAF in one case each. While RAS and BRAF mutations appeared to be mutually exclusive, in one case NF1 mutation was seen co-occurring with MAP2K1 mutation. Protein expression of p-EKR1/2 (Figure 3) was observed in 38/50 (76%) of tumors, with 20 cases scored as 1+, 12 cases as 2+ and 6 cases as 3+ (Figure 4). MAPK pathway gene alterations did not correlate with p-ERK1/2 protein expression (Pearson Chi2, *p* = 0.691). Mutations in MAPK genes and p-ERK1/2 expression were not associated with clinical and survival data.

MAPK pathway activation promoting cell growth, survival and invasion is a frequent event in human cancers, mainly through somatic mutations activating the signal-regulated kinase domain of RAS and BRAF genes of NF1 inactivation, leading to upregulation of ERK1 and ERK2 (Appendix A). Particularly, activating mutations of Ras isoforms or its effectors are found in nearly one-third of all human cancers, especially adenocarcinomas of pancreas, large intestine, biliary tract and small intestine [54]. Previous studies have reported KRAS and HRAS in approximately 15% of ITACs, while BRAF mutations have not been detected [12,13,14,15,16,17,18,19,20,21]. These numbers are consistent with our present NGS study and the one of Sjostedt et al., reporting 6/50 (12%) and 2/19 (11%) KRAS mutation and 1/50 (2%) and 0/19 0%) BRAF mutation, respectively [10]. Our previous NGS study was the first to report inactivating truncating mutations in NF1 in 3/27 cases [31]; in the present series of 50 ITACs we found 4 (8%) NF1 mutated cases. 

In total, 22% of ITACs showed MAPK pathway mutations and could benefit from therapy with selumetinib and other MAPK inhibitors. In addition, KRAS and BRAF mutations are negative predictive factors for anti-EGFR therapies in several tumor types including CRC [55] and, as such, are also important in clinical decision-making.

#### 3.4.4. PI3K Pathway

We identified mutations in PI3K signaling pathway genes in a total of 11 of 50 (22%) cases, among the 29 tumor/germline matched cases 5 in PIK3CA, 1 in PIK3R2 and 1 in MTOR and among the 21 tumor-only cases 4 in TSC2, 1 in MTOR and 1 in AKT1. These mutations occurred in 1 papillary, 6 colonic, 1 solid and 3 mucinous subtypes of ITAC. All mutations were missense except PIK3R2, which was a splice mutation. MTOR and AKT3 each showed a copy number gain in one case. Forty-four of 50 (88%) cases demonstrated immunohistochemical staining of p-mTOR (Figure 3), 18 cases scored as 1+, 16 cases as 2+, and 6 cases as 3+ (Figure 4). Cases with PI3K pathway gene alterations did not show more p-mTOR immunopositivity than cases with no mutations (Pearson chi2, *p* = 0.903). 

PI3K is one of the most frequently dysregulated pathways in human cancers, mostly by PIK3CA, PIK3R1, PTEN, AKT, TSC1, TSC2, STK11 and MTOR genetic alterations (Appendix A) [56]. The PIK3CA mutations found in our series are distributed over exons 2–21 and have been reported previously in ICGC/COSMIC databases to occur in breast and colorectal tumors and recently also in ITAC, together with mutations in PIK3R2, AKT1, MTOR and TSC2 [31]. 

A number of PI3K pathway inhibitors are under clinical testing and the mTOR inhibitors temsirolimus and everolimus and the PI3K inhibitors idelalisib and copanlisib have been FDA approved for the treatment of follicular lymphoma [56] and breast OncoKB. A study in HNSCC reported gain-of-function mutations in PIK3CA to cause an increase in AKT activation, which was related to lower radiation sensitivity [57] HNSCC. To summarize, PI3K pathway alterations occur in 22% of ITACs and are relevant targets for specific inhibitors as well as factors that may influence the efficacy of radiotherapy.

#### 3.4.5. Receptor Tyrosine Kinases (RTKs)

For their capacity to activate MAPK and PI3K signaling pathways, we separately analyzed RTK genes. In the 29 tumor/germline matched cases, we found 9 mutations in ERBB3 (3 cases) ERBB2, ERBB4, KIT, PDGFRA, ROS1 and DDR2 (each one case). Among the 21 tumor-only cases, 8 RTK genes were identified as having probably somatic pathogenic mutations: EPHA2 (3 cases), NTRK1 and ERBB2 (2 cases) and PDGFRA, ROS1, MET, FLT3 and CSF1R (each one case). Taking the two series together, 22/50 (44%) ITACs are affected by potentially damaging mutations in RTK genes. These mutations occurred in all ITAC subtypes. Gene copy number gains could only be evaluated in the 29 tumor/germline matched cases. Here, 10 different RTKs (ERBB2, DDR2, NTRK1, ESR1, ROS1, FGFR1, FGFR3, FLT3 and IGF1R) presented gains affecting in total 9 of the 29 (31%) tumors. MET showed the highest level gain with an estimated 20 copies in one case. RTK gene alterations showed no relation to p-ERK and p-mTOR protein expression (Pearson Chi2, *p* = 0.343 and *p* = 0.156 respectively) (Figure 4).

For its histological similarity, some of the RTK alterations known from lung adenocarcinoma and CRC have also been studied in ITAC and in general showed notable differences between the two tumor types. EGFR and ALK are affected by mutations or chromosomal translocations in lung adenocarcinoma [58], but no abnormalities were found in ITAC [10,19,27,59,60] and neither in our present NGS study. FGFR1 gene copy number gains and overexpression occur in approximately 10% of lung adenocarcinoma [61]. Schröck et al. reported FGFR1 gene copy number gains in 20% of sinonasal squamous carcinoma and 5% of undifferentiated carcinoma, but none in sinonasal adenocarcinomas [29]. Our NGS study did not reveal any FGFR1 mutations, but in 3/29 (10%) ITACs we found copy number gains. FGFR1 is considered an actionable drug target. Based on preclinical experiments with PDX models, Bogatyrova et al. demonstrated higher efficacy of FGFR1 inhibitors in tumors with both FGFR1 copy number gain and overexpression [61]. However, clinical trials with pan-FGFR tyrosine kinase inhibitors have not yielded convincing responses in FGFR1-amplified NSCLC patients. MET overexpression is frequent in CRC and is associated with poor prognosis and resistance to anti-EGFR therapies. An in situ hybridization study on 72 ITACs by Projetti et al. revealed frequent low level gains but no high level amplification of MET, and there was no association with MET protein overexpression [28]. High level amplification was also very infrequent in our study, although one of the two cases with gain of MET had an impressive copy number of 20.

Our NGS study interestingly detected a large proportion (44%) of ITACs to harbor RTK alterations, most of which were copy number gains. Hypothetically, all these cases may benefit from specific inhibitors or from drugs targeting downstream MAPK and PI3K pathways, but this remains to be investigated in future studies.

## 4. Conclusions

Despite the fact that some 20% of all cancer patients are diagnosed with a rare tumor [62,63], our knowledge on the underlying genetic abnormalities in individual types of rare cancers lags far behind. Moreover, as little or no preclinical models are available and hardly any patients are enrolled in clinical trials, there is little progression in treatment options and surgical resection with adjuvant radiotherapy remain the standard of clinical management. ITAC is a typical example of a rare cancer with an estimated incidence rate of less than 0.01 patients per 100,000 inhabitants annually [64,65] and with an overall 5- year survival rate of 40–80%, depending on histological subtype and disease stage [1]. 

Our NGS study demonstrated that a total of 72% of ITACs carry genetic alterations in Wnt, DNA damage response, MAPK and PI3K pathways. This means that a large proportion of ITACs might be treated with specific inhibitors of these pathways, in analogy to CRC and other frequent cancers where such therapies are being applied or tested in clinical trials. Although we placed emphasis on these four signalling pathways, we also identified mutations in genes involved in other cellular processes, such as DNA or histone methylation (KMT2A, EZH2, DNMT3A) or cell differentiation (NOTCH and FOX genes). We found no evidence for genetic differences between mucinous and other histological ITAC subgroups, in contrast to what has been suggested in literature [30,59]. Another preliminary conclusion is that ITAC is genetically heterogeneous and does not display specific characterizing gene mutations, as has been shown for other sinonasal cancer subtypes. However, this may be a consequence of the limited number of 120 cancer-related genes analyzed in our study. Finally, as 9 of the 50 ITACs did not carry any gene mutation, future whole exome or genome sequencing studies are needed to obtain a more complete overview of the mutational landscape of ITAC.

## Figures and Tables

**Figure 1 cancers-13-05022-f001:**
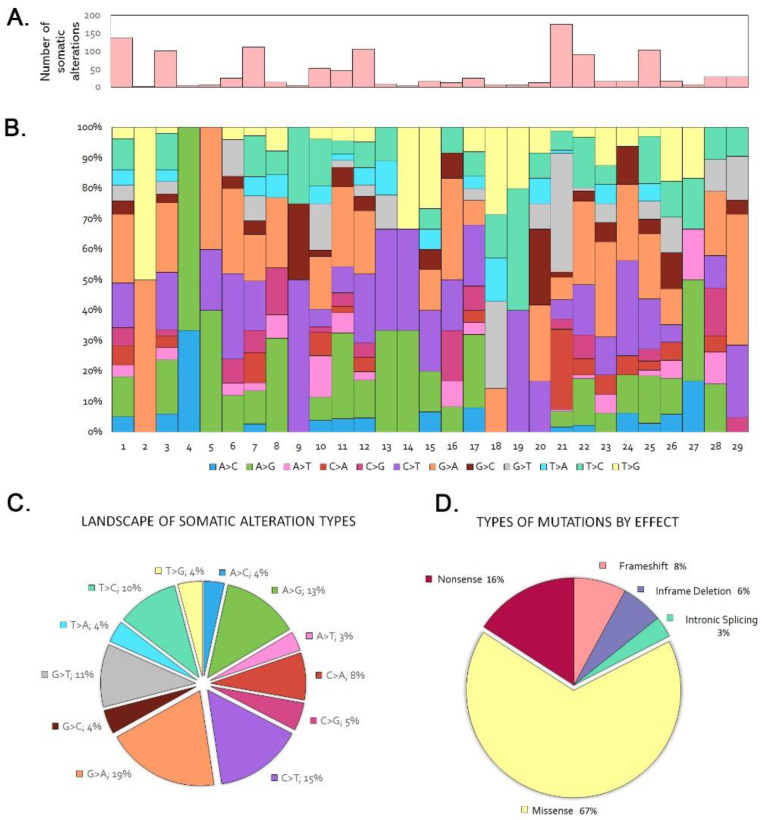
Trends and patterns of somatic and LOH DNA alterations in the sequencing of the 29 tumor/germline matched cases. (**A**) Total number of somatic alterations per tumor. (**B**) Percentage of somatic nucleotide change mutations per tumor. (**C**) Distribution of somatic nucleotide change mutations. (**D**) Distribution of somatic mutation types.

**Figure 2 cancers-13-05022-f002:**
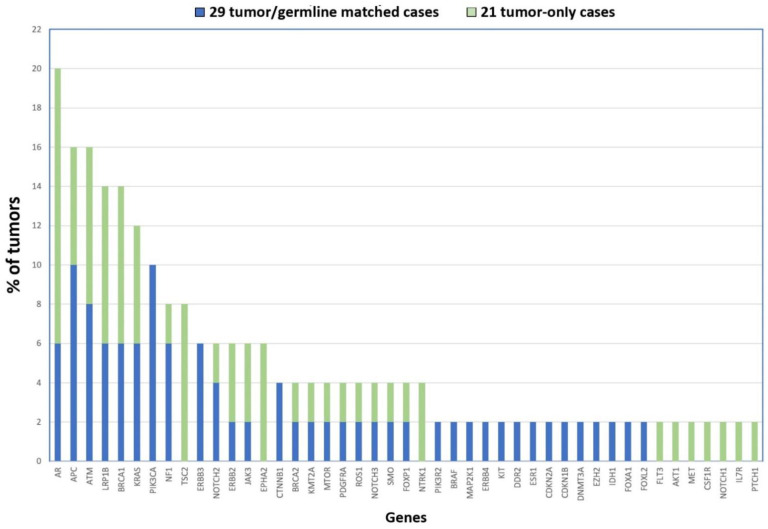
Gene mutations identified in 50 ITACs in order of frequency. The 21 tumor-only cases are shown in blue, and the 29 tumor/germline matched cases are represented in green.

**Figure 3 cancers-13-05022-f003:**
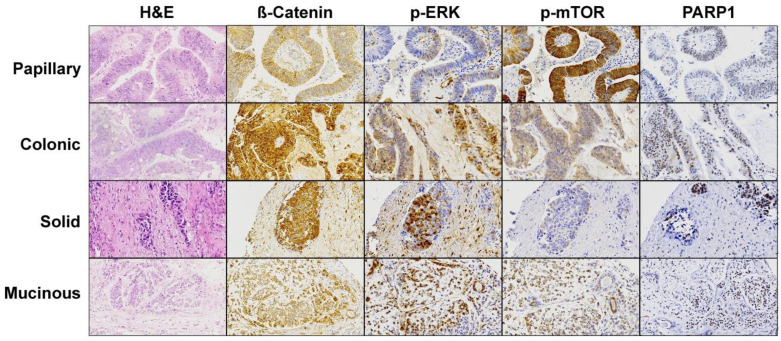
Immunohistochemical staining of the effectors of the most affected pathways found in the cohort of 50 ITAC. A representative image for each histological ITAC subtype is shown. Magnification 20×.

**Figure 4 cancers-13-05022-f004:**
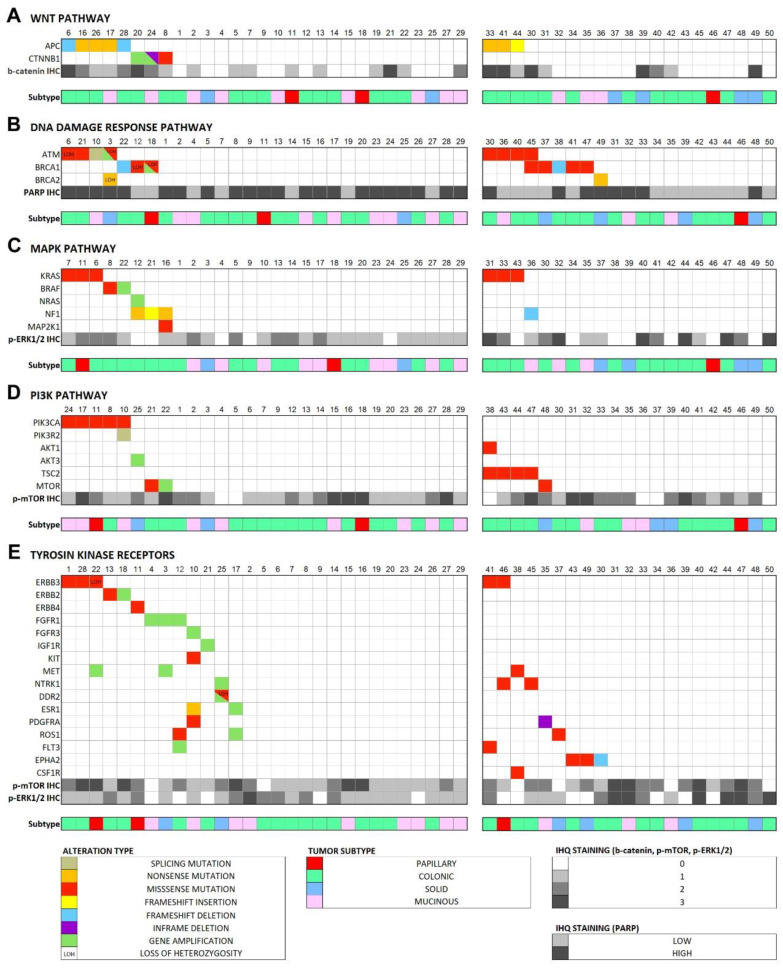
Case-by-case overview of the somatic alterations in the 29 tumor/germline matched cases and the predicted pathogenic alterations in the 21 tumor-only analysis cases, according to affected signaling pathway and immunohistochemical score of the pathway effectors. (**A**) Wnt pathway, (**B**) DNA damage response pathway, (**C**) MAPK pathway, (**D**) PI3K pathway, (**E**) RTK genes.

**Table 1 cancers-13-05022-t001:** Clinical features of 50 ITACs.

Type of Bioinformatic Analysis	T+N	T	Total
**Number of cases**	***n* = 29**	***n* = 21**	***n* = 50**
Age, mean (range) (y)	72 (53–88)	65 (49–82)	69 (49–88)
Gender			
Male	29 (100)	21 (100)	50 (100)
Female	0 (0)	0 (0)	0 (0)
Wood exposure			
Yes	29 (100)	20 (95)	49 (98)
No	0 (0)	1 (5)	1 (2)
Tumor Site			
Maxillary Sinus	0 (0)	0 (0)	0 (0)
Ethmoid Sinus	29 (100)	21 (100)	50 (100)
Histological Subtype			
Papillary	2 (7)	1 (5)	3 (6)
Colonic	16 (55)	13 (62)	29 (58)
Solid	2 (7)	4 (19)	6 (12)
Mucinous	9 (31)	3 (14)	12 (24)
Stage			
I	5 (17)	3 (14)	8 (16)
II	9 (31)	3 (14)	12 (24)
III	10 (35)	7 (34)	17 (34)
IVa	1 (3)	4 (19)	5 (10)
IVb	4 (14)	4 (19)	8 (16)
Patient status			
Alive	12 (41)	2 (9)	14 (28)
DOD	9 (31)	13 (62)	22 (44)
DOC	8 (28)	6 (29)	14 (28)
Locoregional recurrence	17 (59)	14 (67)	31 (62)
Distant metastases	2 (7)	4 (19)	6 (12)
Follow-up, mean (range) (mo)	56 (0–187)	41 (1–153)	50 (0–187)
DFS, mean (range) (mo)	34 (0–107)	28 (0–153)	32 (0–153)

DFS: disease-free survival; DOD: died of disease; DOC: died of other causes; (y): years; (mo): months; T+N: tumor/germline matched cases; T: tumor-only cases.

## Data Availability

Data can be obtained upon reasonable request to the corresponding author.

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
