# Peer review of "Aberrant Signaling Pathways in Sinonasal Intestinal-Type Adenocarcinoma"

_cancers, 2021, doi:10.3390/cancers13195022_

Round 1
Reviewer 1 Report
The authors have sufficiently addressed reviewer queries and requests in this revised version.
Reviewer 2 Report
The authors responded appropriately to the points raised by the reviewer.
This manuscript is a resubmission of an earlier submission. The following is a list of the peer review reports and author responses from that submission.
Round 1
Reviewer 1 Report
The authors conducted next-generation sequencing (NGS) for 120 select genes on 50 sinonasal intestinal-type adenocarcinoma (ITAC) tumor samples, with matching peripheral blood samples for 29 patients. Somatic mutations in select pathways were compared with immunostaining (IHC) of β-Catenin, p-ERK, p-mTOR and PARP1. While the study was limited to sequencing a relatively small number of genes, it focused on cancer-related pathways that are targetable with clinically relevant inhibitor drugs.
Enthusiasm for this manuscript is reduced due to a recently published manuscript on the same topic by this group, and it is unclear how much the samples and analysis overlap between the two studies (Sánchez-Fernández et. al. Next-generation sequencing for identification of actionable gene mutations in intestinal-type sinonasal adenocarcinoma. Sci Rep 11, 2247. 2021). The Introduction and Results/Discussion sections need to include a thorough comparison of the two studies, clearly defining what makes the current study novel.
Other Points:
- The clinical relevance of this data could be improved by correlation of patient overall survival and disease-free survival to the IHC staining for β-Catenin, p-ERK, p-mTOR and PARP1, if this information is available.
- In the Results (page 11), the authors state that several of the mutations were not previously identified in ITAC (e.g. NF1, PIK3CA, PIK3R2, AKT1, MTOR and TSC2). However, the Sánchez-Fernández et. al. publication lists mutations in these same genes, and many of the same mutations are present in both manuscripts. The authors therefore need to specify in the text which particular mutations are new or remove these statements.
- Were any of the gene mutations or categories of mutations associated with tumor stage/grade or patient survival?
- The conclusions and novelty would be strengthened by in vitro inhibition of the identified pathways in proliferation or apoptosis experiments.
- There are a many minor grammatical errors throughout. For example, page 2 line 2: “….especially wood and leather dust, reason why it is considered a professional disease in various European countries.”
Reviewer 2 Report
The paper is pertinent to the Cancers special issue "Sinonasal Cancer: Improving Classification, Stratification and Therapeutic Options". Despite previous papers already described mutations in ITAC, the present manuscript provides a more rational design of the research and identifies novel genes recurrently mutated in this cancer indication providing opportunity to future precision medicine approaches. I recommend few minor revisions before pubblication:
1) I strongly suggest to improve iconography of pathways description with a "pathway map" (eg KEGG or similar).
2) Figure 1: Include a panel with the percentage of mutation type (missense, frameshift, nonsense, in-frame deletions and insertions, and splice mutations).
3) Show figure 2 as percentage over the 21 and 29 ITAC tumors.
4) Figure 4: Letters do not always correspond to the pathway
5) Simplify 3.5 to 3.9 paragraphs
6) Review font and style in 3.6
Reviewer 3 Report
The authors analyzed 120 cancer-related genes in 50 ITACs and investigated the involvement of four specific signaling pathways that are frequently affected by mutations in tumorigenesis using immunohistochemistry. They found gene mutations in the Wnt, DNA damage response, MAPK, and PI3K pathways. ITAC is genetically heterogenous and does not display characteristic gene mutations.
ITAC is a relatively rare neoplasm of the head and neck. This study will contribute to the understanding genetic alterations in this malignancy.
There are some points to reconsider, as described below.
This seems to be an extension of recent work by the authors, In ref 31, the authors reported NGS data of TIACs. It is necessary to clarify which parts/data are overlapped or not in the previous study and the present study.
The purpose of this study is to identify tumorigenic signaling pathways affected by gene mutations and explore their potential therapeutic targets. However, the answer to that goal is not properly stated in the abstract and textual conclusion.
The authors have previously investigated ITAC and CRC and reported high frequency of TP53 mutations in both tumors (ref 10). The authors need to explain why they did not include TP53 in the 120 cancer-related genes in this study.
Based on the signal pathways in which gene mutations were observed, the authors nominated several candidate molecular-targeted drugs for TIAC. If there are similarities in the histology and gene mutations between TIAC and CRC, the molecular-targeted therapies for CRC can be used in the treatment of ITAC. This possibility needs to be explained.
In pages 7-8, the authors stated that “Interestingly, AR gene (Androgen Receptor), is located on chromosome X, supporting the theory that defects in genes on this chromosome could contribute to the male predominance in the development of ITAC apart from the occupational exposure”. Does this mean that mutations in the AR gene result in a defect in AR gene expression in ITAC? A new aspect of this study is the immunohistochemical analysis of ITAC, so the authors can also examine the level of AR expression immunohistologically.
The sections 3.5 to 3.9 can be changed to 3.4.1 to 3.4.5.
Page 2, line 27: Microarray CGH: CGH needs to be spelled out.
Figure 4 legend: (B) should be MAPK pathway, (C) should be PI3K pathway, (D) should be RTK genes, and (E) should be DNA repair genes.
Page 10, line 32: After “–indicating DNA damage response activity”, reference numbers are required to support the relationship between PARP1 expression and ATM, BRCA1, and BRCA2 gene expression.
